# N-acetylcysteine (NAC), an anti-oxidant, does not improve bone mechanical properties in a rat model of progressive chronic kidney disease-mineral bone disorder

Matthew R. Allen [ID][1,2,3,4]*, Joseph Wallace[2], Erin McNerney[1], Jeffry Nyman[5], Keith Avin[3], Neal Chen[3], Sharon Moe[1,3,4]

1 Department of Anatomy, Cell Biology and Physiology, Indiana University School of Medicine, Indianapolis, IN, United States of America, 2 Department of Biomedical Engineering, Indiana University—Purdue University, Indianapolis, IN, United States of America, 3 Division of Nephrology, Department of Medicine, Indiana University School of Medicine, Indianapolis, IN, United States of America, 4 Roudebush VA Medical Center, Indianapolis, IN, United States of America, 5 Department of Orthopaedic Surgery, Vanderbilt University Medical Center, Nashville, TN, United States of America

* matallen@iu.edu

**Data Availability Statement:** All relevant data are within the manuscript and its Supporting Information files.

## Abstract

Individuals with chronic kidney disease have elevated levels of oxidative stress and are at a significantly higher risk of skeletal fracture. Advanced glycation end products (AGEs), which accumulate in bone and compromise mechanical properties, are known to be driven in part by oxidative stress. The goal of this study was to study effects of N-acetylcysteine (NAC) on reducing oxidative stress and improving various bone parameters, most specifically mechanical properties, in an animal model of progressive CKD. Male Cy/+ (CKD) rats and unaffected littermates were untreated (controls) or treated with NAC (80 mg/kg, IP) from 30 to 35 weeks of age. Endpoint measures included serum biochemistries, assessments of systemic oxidative stress, bone morphology, and mechanical properties, and AGE levels in the bone. CKD rats had the expected phenotype that included low kidney function, elevated parathyroid hormone, higher cortical porosity, and compromised mechanical properties. NAC treatment had mixed effects on oxidative stress markers, significantly reducing TBARS (a measure of lipid peroxidation) while not affecting 8-OHdG (a marker of DNA oxidation) levels. AGE levels in the bone were elevated in CKD animals and were reduced with NAC although this did not translate to a benefit in bone mechanical properties. In conclusion, NAC failed to significantly improve bone architecture/geometry/mechanical properties in our rat model of progressive CKD.

## Introduction

Ten percent of Americans suffer from chronic kidney disease (CKD) [1]. Individuals with CKD are at a significantly higher risk of skeletal fracture and associated death, compared to

**Funding:** SM/MA - DK110871, NIH, http://grantome.com/grant/NIH/R01-DK110871-04 The funders had no role in study design, data collection and analysis, decision to publish, or preparation of the manuscript. KA - DK110429 - NIH - http://grantome.com/grant/NIH/K08-DK110429-02. The funders had no role in study design, data collection and analysis, decision to publish, or preparation of the manuscript. SM - I01BX001471 - VA - https://www.research.va.gov/about/funded-proj-details.cfm?pid=545734. The funders had no role in study design, data collection and analysis, decision to publish, or preparation of the manuscript. MA - I01BX003025 - VA - https://www.research.va.gov/about/funded-proj-details.cfm?pid=522036. The funders had no role in study design, data collection and analysis, decision to publish, or preparation of the manuscript.

**Competing interests:** The authors have declared that no competing interests exist.

the normal population [2]. The pathophysiology underling skeletal-related phenotype development in CKD is complex and driven primarily through disturbances in bone and mineral metabolism [3]. Yet there is a growing appreciation for factors outside of mineral metabolism that contribute to skeletal fragility in CKD [4][5].

Advanced glycation end products (AGEs) represent a heterogeneous group of modified proteins and fat that derive from the addition of a carbohydrate to an amino acid or lipid in a non-enzymatic reaction involving sugar. AGE accumulation in the body is influenced by various factors including levels of blood glucose (such as in diabetes), levels of dietary AGE intake, and as both downstream and upstream consequences of increased oxidative stress [6]. Bone represents the largest collection of long-lived proteins in the body, principally collagen, and thus is highly susceptible to AGE accumulation [7]. In vitro [8][9,10], preclinical [11,12], and clinical studies [13] have associated AGE accumulation/collagen cross-linking with reduced bone mechanical properties.

In patients with CKD, regardless of the presence or absence of diabetes, there are increased circulating levels of AGE proteins compared to age-matched controls [14,15]. We and others have found increased skeletal AGE levels in CKD animal models, and we have shown these levels are independent of PTH levels and bone turnover [12][16]. Thus AGE accumulation in bone of patients with CKD may contribute to the poor fracture resistance similar to the reduction in mechanical properties that we have observed in our animal model of CKD [17–19] and impaired bone quality in humans with this disease [20][21][22][23].

The anti-oxidant N-acetylcysteine (NAC) is commonly used in CKD as it has been shown to reduce systemic oxidative stress and AGE formation [24][25]. Whether this translates to reduced levels of AGEs in bone and, most importantly, improved mechanical properties, is unclear. Therefore, the goal of this study was to test the hypothesis that NAC reduces oxidative stress and improves the bone mechanical properties of bone in an animal model of progressive CKD.

## Methods

### Experimental design

Cy/+ rats are characterized by an autosomal dominant mutation in *Anks6*, a gene that codes for the protein SamCystin, and which has been linked to nephronophthisis in humans [26]. In this rat model, the mutation leads to a slow and gradual onset of CKD, paralleling human CKD conditions through the gradual development hyperphosphatemia, hyperparathyroidism, and skeletal abnormalities [27,28].

Male Cy/+ (CKD) rats (n = 25) and unaffected normal littermates (NL; n = 18) were placed on a casein diet (Envigo TD.04539; 0.7% Pi, 0.7% Ca, 15.9% Protein, 5.2% Fat) at 24 weeks of age in order to produce a more consistent kidney disease phenotype [29]. A higher number of CKD animals were started in the study due historical variability in disease and response to treatment. At 30 weeks of age, a subset of CKD (n = 12) and NL (n = 9) rats began treatment with N-acetylcysteine (NAC, 80 mg/kg, IP) for 5 weeks while the remaining animals served as non-treated controls (without vehicle treatment). The 5-week treatment duration was chosen as we have previously shown it to result in dramatic changes to bone properties in this model [18,30]. The NAC dose was based upon previous studies [31][32] and pilot dosing studies in our lab.

At 35 weeks of age, animals were anesthetized with isoflurane and underwent cardiac puncture for blood collection followed by exsanguination and bilateral pneumothorax to ensure death. Tibiae were fixed in 10% neutral buffered formalin and femora were stored in PBS-soaked gauze and frozen at -20C for analysis. All procedures were approved by and carried out

according to the rules and regulations of the Indiana University School of Medicine's Institutional Animal Care and Use Committee.

## Biochemistry

Blood plasma was analyzed for BUN, calcium, and phosphorus using colorimetric assays (Point Scientific, Canton, MI, or Sigma kits). Intact PTH was determined by ELISA (Alpco, Salem, NH). Serum levels of an oxidative stress marker, 8-hydroxy-2' -deoxyguanosine (8-OHdG) were measured using an ELISA kit (Enzo Life Sciences, Farmingdale, NY) and TBARS were measured using TBARS assay (TCA method) kit (Cayman Chemical, Ann Arbor, MI).

## Micro-Computed Tomography (microCT)

Proximal tibia were scanned with microCT (Skyscan 1172) using a 12 micron isotropic voxel size for determination of trabecular bone volume (BV/TV, %) and cortical porosity, using methods previously published [18]. BV/TV was measured within a region of interest in the proximal metaphysis starting at ~0.5 mm below the most distal region of the growth plate and extending downward for 1 mm. The ROI was manually drawn to excluded cortical bone. Cortical porosity was assessed on a single slice at the most distal edge of the trabecular ROI with the isolation of the cortex achieved through manual segmentation. Whole femur were scanned (Skyscan 1176) at a nominal resolution of 18 microns to assess geometric properties including total tissue area (TA), bone area (BA), polar cross-sectional moment of inertia (CSMIp) and cortical thickness (Ct.Th) on a single slice located at 50% of bone length [33]. Scans and analyses were done in accordance with standard guidelines [34].

## Whole bone mechanics

Structural mechanical properties of the left femur were determined by four-point bending as previously described [35]. The anterior surface was placed on two lower supports located ±9 mm from the mid-diaphysis (18 mm span length) with an upper span length of 6 mm. Specimens were loaded to failure at a rate of 2 mm/min, producing a force-displacement curve for each sample. Structural-dependent mechanical properties were obtained directly from these curves, while apparent material properties were derived from the force-displacement curves, CSMIp from microCT, and the distances from the centroid to the tensile surface using standard beam-bending equations for four-point bending [36].

## Measurements of skeletal collagen cross-linking of femoral cortical bone

After mechanical testing, segments of the femoral cortex (~3 mm in length) were processed for assessment of enzymatic collagen crosslinks and an AGE crosslink (pentosidine) by a high performance liquid chromatography (HPLC) system (Beckman-Coulter System Gold 168) as previously published [12,37,38]. Briefly, the femoral shaft was flushed with saline to remove marrow followed by incubating with Immunocal (Decal Chemical Corporation, Congers, NY) to demineralize the bone for 2–3 days. Demineralization end-point determination assays (Polysciences, Warrington, PA) were used to verify demineralization of each specimen. The demineralized bone tissues were then hydrolyzed in 6 M HCL at 110˚C for 16 hours. Standards of pyridinoline (PYD; Quidel), deoxypyridinoline (DPD; Quidel), and pentosidine (PE; International Maillard Reaction Society) were used. Hydroxyproline levels, also measured by HPLC, were used to normalize crosslink concentration (mol/mol collagen).

## Measurement of total AGEs in bone tissue

The same demineralized bone tissues that were hydrolyzed for enzymatic cross-links were assessed for total AGE content using previously published methods [10,39]. Briefly, fluorescence readings taken with a CLARIOstar high performance microplate reader (BMG LAB-TECH Inc, Cary, NC) at wavelengths of 370nm/440nm excitation/emission against a quinine sulfate standard and normalized by the collagen content for the sample. The amount of collagen for each bone specimen was based on the amount of hydroxyproline determined by a hydroxyproline assay kit according to manufactures' instruction (Sigma-Aldrich USA, St. Louis, MO).

## Statistics

Comparisons among groups were assessed by two-way ANOVA with the main effects being disease (NL and CKD) and treatment (control vs. NAC). When a significant interaction was found in the two-way ANOVA Fisher's LSD post-hoc tests across all four groups were performed. *A priori* α-levels were set at 0.05 to determine statistical significance. Data are presented as mean ± standard deviation.

## Results

### The effect of NAC on kidney function, mineral metabolism and oxidative stress in CKD rats

CKD animals had significantly higher BUN (4-fold), phosphorus (1.7-fold) and serum PTH (8-fold) compared to NL (**Table 1**). NAC did not reduce PTH or BUN but did result in significantly lower phosphorus and significantly higher calcium in in CKD animals. There was no difference in levels of serum oxidative stress markers 8-OHdG (**Fig 1A**) or TBARS (**Fig 1B**) between NL and CKD rats. NAC had no effect on levels of 8-OHdG but decreased levels of TBARS (**Fig 1A & 1B**).

### The effect of NAC on skeletal morphology of femoral mid-diaphysis in CKD rats

There was no significant effect of phenotype, treatment, or an interaction on proximal tibia trabecular BV/TV (**Fig 2A**). Cortical porosity showed a significant main effect of phenotype (higher in CKD), but it was not affected by treatment (**Fig 2B**). At the mid-diaphysis, bone area, cross-sectional moment of inertia, and cortical thickness were all significantly affected by genotype (lower in CKD) without being significantly affected by NAC. Only CKD animals had

**Table 1. Systemic biochemical markers.**

|  | Normal | | CKD | | Main effect of disease | Main effect of treatment | Disease x treatment interaction |
|---|---|---|---|---|---|---|---|
|  | Control (n = 9) | NAC (n = 9) | Control (n = 13) | NAC (n = 12) | | | |
| Calcium, mg/dL | 6.8 ± 1.2 | 7.3 ± 1.2 | 5.6 ± 1.9 | 8.0 ± 1.7 * | 0.628 | **0.004** | **0.048** |
| Phosphorous, mg/dL | 5.1 ± 1.1 | 5.8 ± 2.8 | 8.8 ± 3.3 | 7.6 ± 3.3 | **0.004** | 0.808 | 0.281 |
| BUN, mg/dL | 14.1 ± 2.4 | 14.7 ± 2.9 | 57.8 ± 12.2 | 54.4 ± 12.5 | **<0.001** | 0.650 | 0.513 |
| PTH, pg/mL | 190 ± 64 | - | 1499 ± 1117 | 994 ± 746 | **<0.001** | 0.201 | NA |

Data presented as mean and standard deviation. CKD, chronic kidney disease; NAC, N-acetylcysteine; BUN, blood urea nitrogen; PTH, parathyroid hormone.

*p< 0.05 versus control within disease.

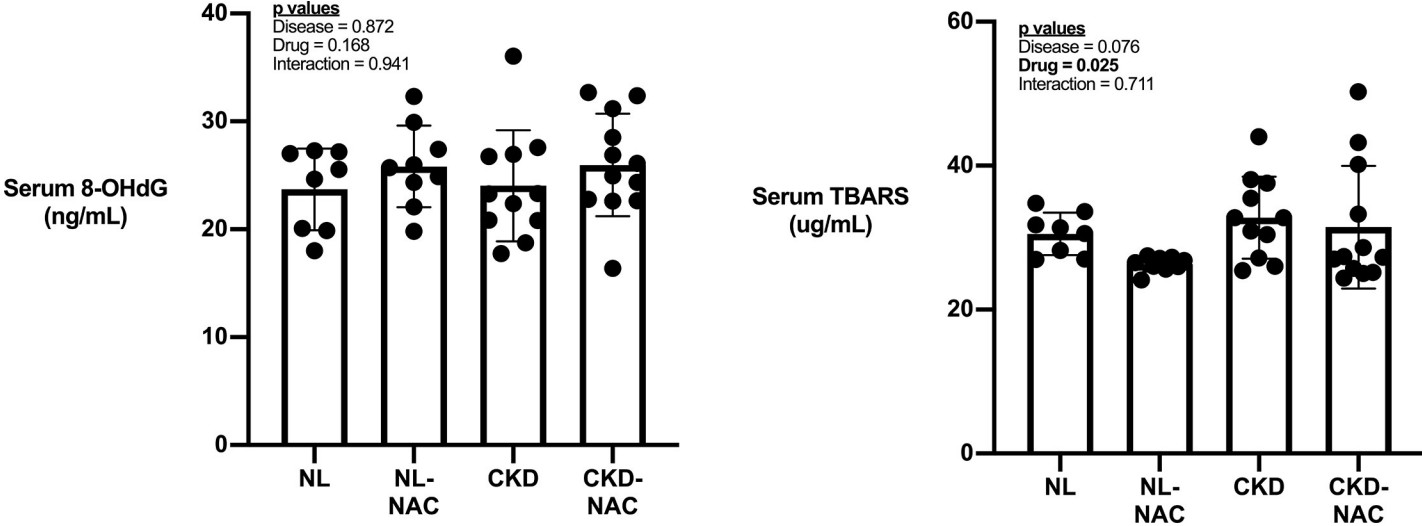

**Fig 1. The systemic marker of oxidative stress.** Serum 8-OHdG (A) and TBPARS (B) in NL and CKD rats treated with or without NAC. Data presented as mean and standard deviation with point plots of individual data. (n = 8–12 each group).

measurable cortical porosity at the mid-diaphysis and levels were not affected by NAC treatment (**Table 2**).

## The effect of NAC on bone mechanics in CKD rats

The majority of mechanical properties, both structural and estimated material-level, exhibited significant main effects of genotype, with CKD animals having inferior properties compared to NL (**Table 3**). There was no main effect of NAC treatment nor an interaction between genotype and treatment.

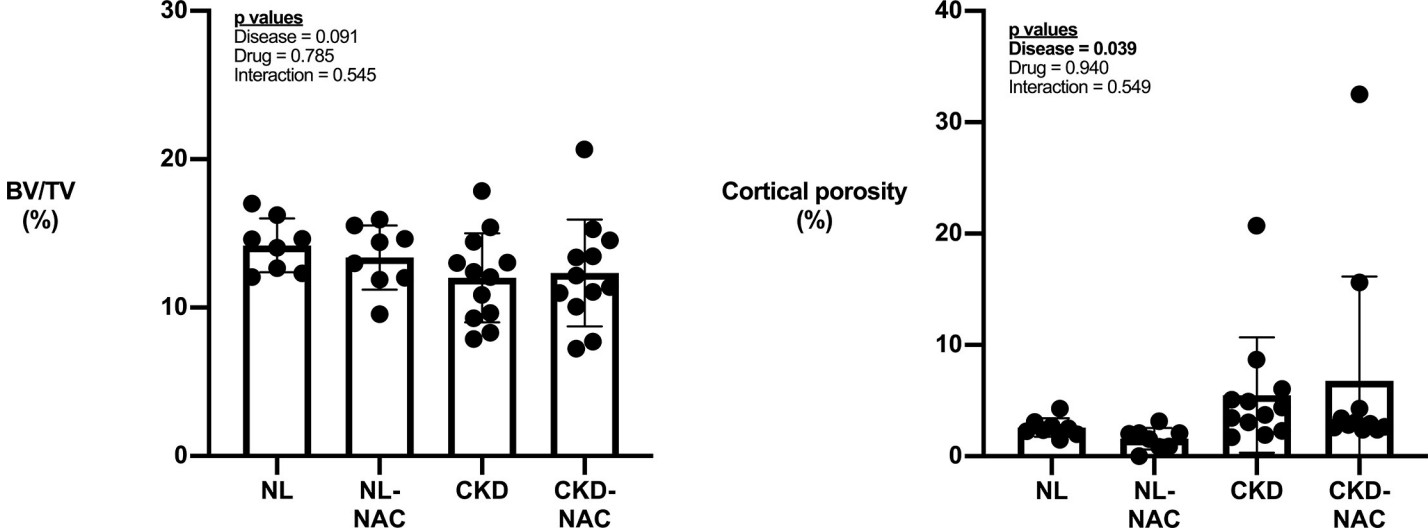

**Fig 2. The effect of CKD and NAC on bone architecture.** Cortical, but not trabecular bone is affected by CKD with no effect of NAC on either compartment. A) Proximal tibia trabecular bone was not significantly affected by NAC in either genotype. B) Cortical porosity was significantly higher in CKD animals while NAC did not affect levels in either genotype. Data presented as mean and standard deviation with point plots of individual data. (n = 8–12 each group).

**Table 2. Skeletal morphology of femoral mid-diaphysis.**

| | Normal | | CKD | | Main effect of disease | Main effect of treatment | Disease x treatment interaction |
|---|---|---|---|---|---|---|---|
| | Control (n = 8) | NAC (n = 8) | Control (n = 11) | NAC (n = 11) | | | |
| Bone area, mm$^2$ | 8.8 ± 0.8 | 9.1 ± 0.5 | 7.9 ± 1.0 | 7.5 ± 0.8 | **0.001** | 0.975 | 0.194 |
| Cross-sectional moment of inertia, mm$^4$ | 9.9 ± 1.2 | 10.9 ± 1.3 | 9.4 ± 1.5 | 8.7 ± 1.1 | **0.003** | 0.701 | 0.071 |
| Cortical thickness, mm | 0.74 ± 0.04 | 0.77 ± 0.04 | 0.63 ± 0.09 | 0.61 ± 0.13 | **0.001** | 0.876 | 0.463 |

Data presented as mean and standard deviation. CKD, chronic kidney disease; NAC, N-acetylcysteine.

## The effect of NAC on total bone AGE accumulation and skeletal collagen cross-linking of femoral cortical bone

Total AGEs in femoral bone tissue from untreated CKD rats were significantly higher compared to normal animals (**Fig 3**). There was a significant interaction for total bone AGE with NAC-treated CKD rats having lower total bone AGE levels whereas NL rats treated with NAC had no effect compared to untreated NL rats (**Fig 3**). Assessment of enzymatic collagen cross-links and pentosidine by HPLC revealed that pyridinoline levels per unit collagen were significantly lower in CKD animals compared to NL with no effect of NAC treatment (**Table 4**). There was a significant interaction for pentosidine levels with NAC-treated NL animals having lower skeletal pentosidine levels while CKD animals treated with NAC were not significantly different compared to untreated controls (**Table 4**). Total AGE levels were negatively correlated to a number of mechanical properties with r values of ~0.3 that were statistically signfiicant (**Table 5**).

## Discussion

The systemic disruptions in mineral metabolism associated with chronic kidney disease are profound and result in pathophysiological changes in numerous organ systems including skeletal changes and calcification of arteries. Oxidative stress is common in many diseases including chronic kidney disease [40]. Although the cause is often multi-factorial, reducing oxidative stress represents a general approach to lessen the consequence of disease. In the present study we demonstrated that NAC, an anti-oxidant, given beginning at 30 weeks of age

**Table 3. Mechanical properties of the femur mid-diaphysis.**

| | Normal | | CKD | | Main effect of disease | Main effect of treatment | Disease x treatment interaction |
|---|---|---|---|---|---|---|---|
| | Control (n = 8) | NAC (n = 7) | Control (n = 12) | NAC (n = 10) | | | |
| Ultimate load, N | 261 ± 19 | 252 ± 23 | 184 ± 45 | 183 ± 45 | **<0.001** | 0.667 | 0.739 |
| Stiffness, N/mm | 441 ± 36 | 415 ± 48 | 364 ± 42 | 347 ± 38 | **<0.001** | 0.125 | 0.755 |
| Total displacement, μm | 846 ± 101 | 883 ± 128 | 662 ± 137 | 768 ± 187 | **0.003** | 0.207 | 0.380 |
| Work to failure, Nmm | 129 ± 22 | 120 ± 24 | 70 ± 33 | 80 ± 39 | **<0.001** | 0.972 | 0.381 |
| Ultimate stress, MPa | 150 ± 17 | 140 ± 22 | 110 ± 32 | 114 ± 26 | **<0.001** | 0.706 | 0.419 |
| Modulus, GPa | 7.2 ± 0.8 | 6.5 ± 0.8 | 6.3 ± 1.0 | 6.4 ± 0.6 | 0.065 | 0.258 | 0.175 |
| Total strain, μE | 30371 ± 4115 | 31088 ± 4297 | 22967 ± 5242 | 26338 ± 6739 | **0.002** | 0.263 | 0.465 |
| Toughness, MPa | 2.59 ± 0.54 | 2.34 ± 0.55 | 1.45 ± 0.74 | 1.70 ± 0.84 | **0.001** | 0.989 | 0.297 |

Data presented as mean and standard deviation. CKD, chronic kidney disease; NAC, N-acetylcysteine.

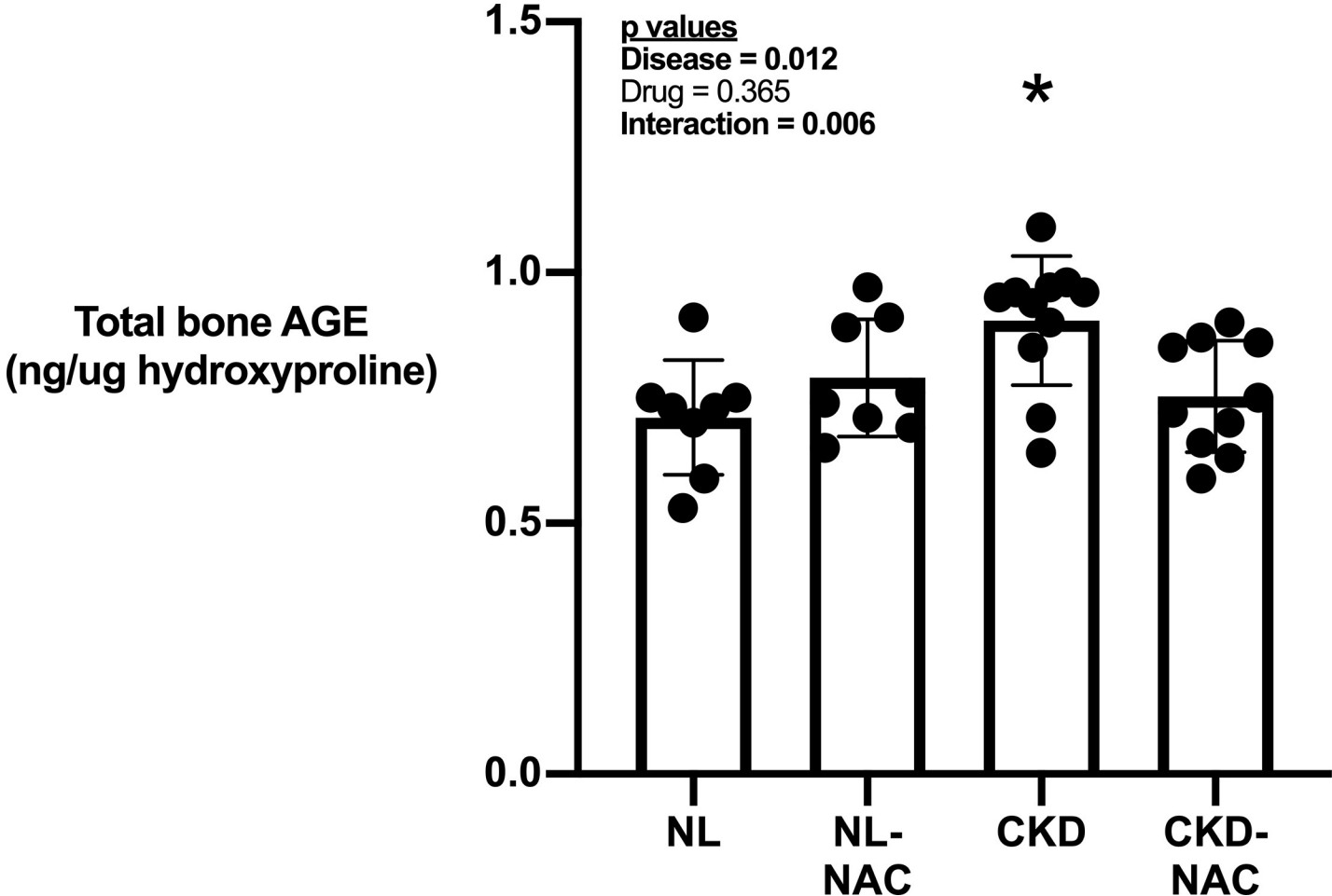

**Fig 3. The effect of CKD and NAC on AGE accumulation.** There was increased accumulation of AGE in femoral shaft from CKD rats compared to control rats (Disease effect). There is no treatment effect overall but there is a significant interaction in which treatment with NAC reduced bone AGE accumulation in CKD rats but not in NL rats. Data presented as mean and standard deviation with point plots of individual data (n = 8–12 each group). *p < 0.05 vs all other groups.

(approximately 60–70% reduction in kidney function) for 5 weeks did not alter kidney function, assessed by BUN, or mineral metabolism in CKD rats. While NAC marginally lowered serum levels of TBARS, a marker of lipid oxidation, it had no significant effect on serum

**Table 4. Skeletal collagen cross-linking of femoral cortical bone.**

| | Normal | | CKD | | Main effect of disease | Main effect of treatment | Disease x treatment interaction |
|---|---|---|---|---|---|---|---|
| | Control (n = 8) | NAC (n = 9) | Control (n = 13) | NAC (n = 12) | | | |
| Pyridinoline/mmol collagen | 0.195 ± 0.03 | 0.20 ± 0.02 | 0.163 ± 0.02 | 0.168 ± 0.03 | **0.001** | 0.679 | 0.817 |
| Deoxypyridinoline/mmol collagen | 0.131 ± 0.02 | 0.127 ± 0.02 | 0.119 ± 0.03 | 0.114 ± 0.03 | 0.138 | 0.595 | 0.976 |
| Pentosidine/mmol collagen | 70 ± 27 | 53 ± 10 * | 59 ± 14 | 75 ± 31 | 0.444 | 0.925 | **0.023** |

Data presented as mean and standard deviation. CKD, chronic kidney disease; NAC, N-acetylcysteine.

*p< 0.05 versus control within disease.

**Table 5. Correlation matrix between oxidative stress/collagen parameters and bone geometry/mechanics.**

| | Ultimate Force | Total Displacement | Stiffness | Total Work | Ultimate Stress | Total Strain | Modulus | Toughness | Bone area | CSMI | Cortical thickness |
|---|---|---|---|---|---|---|---|---|---|---|---|
| **Total bone AGE** | -0.360 | -0.335 | -0.288 | -0.368 | -0.298 | -0.353 | -0.103 | -0.347 | -0.376 | -0.223 | -0.478 |
| **8-OHdG** | -0.388 | -0.316 | -0.364 | -0.335 | -0.357 | -0.335 | -0.177 | -0.347 | -0.025 | -0.040 | -0.134 |
| **PYD per collagen** | 0.280 | 0.144 | 0.340 | 0.196 | 0.232 | 0.139 | 0.254 | 0.165 | 0.255 | 0.038 | 0.336 |
| **DPD per collagen** | 0.324 | 0.324 | 0.194 | 0.297 | 0.361 | 0.339 | 0.202 | 0.328 | 0.248 | 0.243 | 0.290 |
| **PE per collagen** | -0.220 | -0.168 | -0.131 | -0.148 | -0.098 | -0.155 | 0.110 | -0.079 | -0.325 | -0.335 | -0.272 |

Data presented as r values with red highlighted cells noting correlations with $p < 0.05$. AGE, advanced glycation end products; PYD, Pyridinoline; DPD, Deoxypyridinoline; PE, Pentosidine; CSMI, cross-sectional moment of inertia

8-OHDG, a marker of DNA oxidation. Furthermore, we found no end-organ effects of NAC on the skeleton in CKD animals.

A major negative finding of this work is the absence of systemic oxidative stress elevations in the CKD animals and modulation of levels in either treatment. We chose to assess oxidative stress using 8-OHDG as it has shown to be elevated in our model of progressive chronic kidney disease [41] and can be modulated by NAC [40]. Surprisingly, not only was there no effect of disease, but there was no effect of NAC in either normal or disease conditions. Conversely, there was a significant effect of NAC on lowering of TBARS, a common assessment of lipid peroxidation. Previous work has shown that 7 weeks of NAC treatment (dosed orally at 150 mg/kg/day) reduced sodium fluoride-induced oxidative stress, as assessed by 8-OHDG [42]. Whether our conflicting results are due to different etiologies of oxidative stress, the need for additional measures of oxidative stress, different dosing routes (or doses), or treatment time are not clear. It is also unclear why there was no disease effect on oxidative stress given we have seen this previously in our animals. This could be due to the heterogeneity in disease severity we see across cohorts, although we observed many disease-induced changes that are typical in our model, such as changes to BUN, PTH, and bone outcomes.

The influence of oxidative stress on bone has been studied predominantly in the context of signaling bone cells. A lesser appreciated effect of oxidative stress is the effects it has on the existing matrix. AGE accumulation in bone is associated with negative effects on bone mechanical properties [7]. Their accumulation has been proposed to be responsible for the bone fragility associated with diabetes, a result of high blood sugar and circulating AGE levels (measured as HbA1c). Here we used a rat model of progressive CKD and demonstrated that total bone AGE, but not pentosidine levels, are increased in CKD rats compared to NL. Previous studies in this same CKD model have documented both higher [12] and unchanged [38] pentosidine levels in CKD animals. However, our current work documents that NAC lowered total bone AGE levels, with no effect on bone pentosidine levels in in CKD animals. It is well-known that pentosidine is just one of many AGEs within bone and that it's concentration is relatively low compared to some others [43] Given the link between AGEs and bone is in part through their effects on mechanical properties [8][44][45], it is important to note that we found there was no appreciable benefit of NAC on any mechanical properties, despite CKD inducing many of the expected effects on reducing mechanical properties. This certainly points to the concept that AGEs are not the whole story and that the metabolic disturbances in CKD are having other tissue-level effects.

In addition to those discussed above, this work is limited in that assessments of the degree of redox in the bone were not assessed. Therefore it is possible that changes (or lack of change) in the serum does not reflect what is happening in the bone tissue.

In summary, NAC failed to alter the skeletal abnormality of our rat model of progressive CKD. While it is plausible that higher doses or longer treatment may have had more benefit, it is more likely that the pathophysiology of bone in CKD-MBD are multifactorial in etiology and that oxidative stress treatment is insufficient to prevent negative skeletal effects.

## Supporting information

**S1 Data.**
(XLSX)

**S2 Data.**
(XLSX)

**S3 Data.**
(XLSX)

**S4 Data.**
(XLSX)

## Author Contributions

**Conceptualization:** Matthew R. Allen, Joseph Wallace, Keith Avin, Neal Chen, Sharon Moe.

**Formal analysis:** Matthew R. Allen, Erin McNerney, Jeffry Nyman, Keith Avin, Neal Chen.

**Funding acquisition:** Matthew R. Allen, Sharon Moe.

**Methodology:** Joseph Wallace, Jeffry Nyman, Keith Avin.

**Project administration:** Matthew R. Allen, Sharon Moe.

**Supervision:** Matthew R. Allen, Neal Chen, Sharon Moe.

**Writing – original draft:** Matthew R. Allen, Sharon Moe.

**Writing – review & editing:** Matthew R. Allen, Joseph Wallace, Erin McNerney, Jeffry Nyman, Keith Avin, Neal Chen, Sharon Moe.

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
