## [Decision Letter · Decision Letter 0]

9 Jan 2020

PONE-D-19-34955

N-acetylcysteine (NAC), an anti-oxidant, does not improve bone mechanical properties in a rat model of progressive chronic kidney disease-mineral bone disorder

PLOS ONE

Dear Dr. Allen,

Thank you for submitting your manuscript to PLOS ONE. After careful consideration, we feel that it has merit but does not fully meet PLOS ONE’s publication criteria as it currently stands. Therefore, we invite you to submit a revised version of the manuscript that addresses the points raised during the review process.

Two reviewers have evaluated your manuscript. Fortunately, both found it of some interest. However, there are major issues that need to be resolved. Please address ALL their comments in your rebuttal and revised manuscript. 

It is essential that data on CKD status are provided. At least plasma or serum creatinine and urea (more reliable than creatinine in rodents) and, if 24h urine is available, creatinine clearance. With 24h urine you can also measure TBARS excretion in the urine. This is often a useful integrative measure of oxidative damage.

Additional animal experiments are probably not necessary.

To enhance the reproducibility of your results, we recommend that if applicable you deposit your laboratory protocols in protocols.io, where a protocol can be assigned its own identifier (DOI) such that it can be cited independently in the future. For instructions see: http://journals.plos.org/plosone/s/submission-guidelines#loc-laboratory-protocols

We look forward to receiving your revised manuscript.

Kind regards,

Jaap A. Joles, DVM, PhD

Academic Editor

PLOS ONE

Journal Requirements:

Reviewers' comments:

Reviewer's Responses to Questions

**Comments to the Author**

1. Is the manuscript technically sound, and do the data support the conclusions?

Reviewer #1: Partly

Reviewer #2: Yes

2. Has the statistical analysis been performed appropriately and rigorously? 

Reviewer #1: Yes

Reviewer #2: Yes

3. Have the authors made all data underlying the findings in their manuscript fully available?

Reviewer #1: Yes

Reviewer #2: Yes

4. Is the manuscript presented in an intelligible fashion and written in standard English?

Reviewer #1: Yes

Reviewer #2: Yes

5. Review Comments to the Author

Reviewer #1: The authors reported the effect of N-acetylcysteine (NAC) treatment on systemic oxidative stress, the degree of bone collagen crosslinks, and bone mechanical properties in progressive chronic kidney disease (CKD) model rats. This study is important to understand the role of oxidative stress for deterioration of bone mechanical properties in CKD. There are some concerns as follows;

1) It is well known that the level of systemic or local oxidative stress increases in CKD condition. The previous study of the authors also reported 8-OHdG in serum was increased in 35weeks old Cy/+ rats (Avin KG, et al PLOS ONE 2016). However, why were the levels of oxidative stress marker in serum similar between Normal and CKD rats in this study? The authors should discuss this point.

2) Does the degree of redox in bone tissue reflect on the degree of redox in serum? Did the authors determine the level of anti-oxidant markers such as superoxide dismutase, or glutathione in both bone tissue and serum? It is necessary for the authors to present these data or argue this point while using previous studies.

3) What does the reducing of bone mechanical properties in CKD rats depend on in this study? Does the amount of total bone AGEs, or pentosidine correlate with bone mechanical properties?

4) What the level of serum creatinine or creatinine clearance in their CKD rats? This data is important to understand CKD related bone abnormalities. The authors should describe this data.

5) Was the method of total bone AGEs assessed by microplate reader authorized in this case? Did the value obtained by this method reflect AGEs content? The authors should address much more in detail about this method.

6) The authors described calcium and phosphorus content in materials and methods. It is well known that protein and fat content in diet affects kidney function and bone health. It is better for readers to have this information.

7) In the discussion, the authors described no effect of NAC on the aorta in CKD animals. Which data was this shown by?

8) In Table 4, was the data of pentosidine content in the two CKD groups correct? Was this data not replaced? It is recommended to check this data.

Reviewer #2: 1. The format of the tables should be modified to standard three-line table or four-line table and the results of pairwise comparisons should be indicated used the symbols, like “*”or others.

2. In the figures, the results of pairwise comparisons should also be indicated used the symbols and horizontal lines, which indicated the difference occurred in indicated two groups.

3. In the part of “The effect of NAC on total bone AGE accumulation and skeletal collagen cross-linking of femoral cortical bone”, the statement of ‘Assessment of enzymatic collagen crosslinks and pentosidine by HPLC revealed that pyridinoline levels per unit collagen were significantly lower in CKD animals compared to NL with no effect of NAC treatment (Table 4).’ is not clear.

4. In the ‘Statistics’ part, the statement should be clearer and should tell the readers the meaning of “interaction”.

5. The authors could make some bivariate correlation analysis, such as the correlation analysis of oxidative stress indicators with the AGEs, or the correlation relationship of AGEs with bone indicators of micro CT and bone biomechanics.

6. The expression and grammar should be improved, errors about singular and plural should be corrected.

6. PLOS authors have the option to publish the peer review history of their article (what does this mean?). If published, this will include your full peer review and any attached files.

Reviewer #1: No

Reviewer #2: No

---

## [Author Response · Author response to Decision Letter 0]

20 Feb 2020

Reviewer #1: 

1) It is well known that the level of systemic or local oxidative stress increases in CKD condition. The previous study of the authors also reported 8-OHdG in serum was increased in 35weeks old Cy/+ rats (Avin KG, et al PLOS ONE 2016). However, why were the levels of oxidative stress marker in serum similar between Normal and CKD rats in this study? The authors should discuss this point.

The reviewer is certainly correct that we and others have noted elevated oxidative stress in CKD. We in fact have seen altered oxidative stress in two papers using this animal model. We are unclear as to the reasons they were not elevated here, as that was certainly the basis for the study/intervention with NAC. We have tried to make this explicitly clear in the discussion. Unfortunately, we don’t have great hypotheses for this finding (or lack of finding).

2) Does the degree of redox in bone tissue reflect on the degree of redox in serum? Did the authors determine the level of anti-oxidant markers such as superoxide dismutase, or glutathione in both bone tissue and serum? It is necessary for the authors to present these data or argue this point while using previous studies.

The reviewer raises a great question about relation of serum and bone levels but unfortunately we are not able to make those assessments in both bone and serum. We have added this as a limitation of the work.

3) What does the reducing of bone mechanical properties in CKD rats depend on in this study? Does the amount of total bone AGEs, or pentosidine correlate with bone mechanical properties?

We have added a correlation matrix table and a limited amount of text to the discussion. Total bone AGEs and 8-OHdG are both are negatively correlated to mechanical properties (r values around -0.3 for many variables). These relationships are not overly strong, albeit statistically significant. 

4) What the level of serum creatinine or creatinine clearance in their CKD rats? This data is important to understand CKD related bone abnormalities. The authors should describe this data.

We did not measure serum or urine creatinine in the current study. We have previously undertaken creatine clearance in our animal model and shown that they correlated to BUN. in our animal model. Thus we are confident that BUN accurately reflects that CKD animals in this study had kidney disease and that NAC did not affect the degree of altered kidney function.

5) Was the method of total bone AGEs assessed by microplate reader authorized in this case? Did the value obtained by this method reflect AGEs content? The authors should address much more in detail about this method.

We followed previously published methods for measure of total AGEs. We have added citations to the methods section.

6) The authors described calcium and phosphorus content in materials and methods. It is well known that protein and fat content in diet affects kidney function and bone health. It is better for readers to have this information.

We have included dietary values of protein (17.7%) and fat (5.2%) in the methods..

7) In the discussion, the authors described no effect of NAC on the aorta in CKD animals. Which data was this shown by?

We have removed reference to the aorta – sorry for the confusion.

8) In Table 4, was the data of pentosidine content in the two CKD groups correct? Was this data not replaced? It is recommended to check this data.

We have verified that the pentosidine data in Table 4 is correct.

Reviewer #2: 

1. The format of the tables should be modified to standard three-line table or four-line table and the results of pairwise comparisons should be indicated used the symbols, like “*”or others.

We are happy to edit the table as needed for publication but are unsure what the reviewer is requesting here.

2. In the figures, the results of pairwise comparisons should also be indicated used the symbols and horizontal lines, which indicated the difference occurred in indicated two groups.

We have revised our figures to make it clear that the notations in the upper left corner are main effect and interaction p values. In figures 1B and 2B, where there are main effects, we find it more complicated to add specific notations to the figure – these main effects make it clear that either drug or disease are different. For Figure 3 we have a notation in the figure legend that the * symbol represents difference from all other groups. We hope this is clear.

3. In the part of “The effect of NAC on total bone AGE accumulation and skeletal collagen cross-linking of femoral cortical bone”, the statement of ‘Assessment of enzymatic collagen crosslinks and pentosidine by HPLC revealed that pyridinoline levels per unit collagen were significantly lower in CKD animals compared to NL with no effect of NAC treatment (Table 4).’ is not clear.

This statement matches the table, which shows a significant main effect of disease for pyridinoline (0.0001) but not for treatment (and no interaction). 

4. In the ‘Statistics’ part, the statement should be clearer and should tell the readers the meaning of “interaction”.

We have clarified the results section and the tables.

5. The authors could make some bivariate correlation analysis, such as the correlation analysis of oxidative stress indicators with the AGEs, or the correlation relationship of AGEs with bone indicators of micro CT and bone biomechanics.

We have added a correlation matrix table to the paper (Table 5) and referenced it in the results section.

6. The expression and grammar should be improved, errors about singular and plural should be corrected.

We have tried to correct grammar where we saw inconsistency.

---

## [Decision Letter · Decision Letter 1]

28 Feb 2020

N-acetylcysteine (NAC), an anti-oxidant, does not improve bone mechanical properties in a rat model of progressive chronic kidney disease-mineral bone disorder

PONE-D-19-34955R1

Dear Dr. Allen,

We are pleased to inform you that your manuscript has been judged scientifically suitable for publication and will be formally accepted for publication once it complies with all outstanding technical requirements.

With kind regards,

Jaap A. Joles, DVM, PhD

Academic Editor

PLOS ONE

Additional Editor Comments (optional):

Reviewers' comments:

Reviewer's Responses to Questions

**Comments to the Author**

1. If the authors have adequately addressed your comments raised in a previous round of review and you feel that this manuscript is now acceptable for publication, you may indicate that here to bypass the “Comments to the Author” section, enter your conflict of interest statement in the “Confidential to Editor” section, and submit your "Accept" recommendation.

Reviewer #1: All comments have been addressed

Reviewer #2: All comments have been addressed

2. Is the manuscript technically sound, and do the data support the conclusions?

Reviewer #1: Yes

Reviewer #2: Yes

3. Has the statistical analysis been performed appropriately and rigorously? 

Reviewer #1: Yes

Reviewer #2: Yes

4. Have the authors made all data underlying the findings in their manuscript fully available?

Reviewer #1: Yes

Reviewer #2: Yes

5. Is the manuscript presented in an intelligible fashion and written in standard English?

Reviewer #1: Yes

Reviewer #2: Yes

6. Review Comments to the Author

Reviewer #1: The authors have revised their article due to the reviewer’s comments and resubmitted it. The major concerns that were pointed by reviewers have been revised. Their study is important to argue the role of pentosidine crosslinks, total advanced glycation end products, and oxidative stress on bone mechanical properties in CKD.

Reviewer #2: the authors had answered solved all queries from reviewers.

at this stage, I have no more suggestion.

7. PLOS authors have the option to publish the peer review history of their article (what does this mean?). If published, this will include your full peer review and any attached files.

Reviewer #1: No

Reviewer #2: No

---

## [Editor Report · Acceptance letter]

9 Mar 2020

PONE-D-19-34955R1 

N-acetylcysteine (NAC), an anti-oxidant, does not improve bone mechanical properties in a rat model of progressive chronic kidney disease-mineral bone disorder 

Dear Dr. Allen:

I am pleased to inform you that your manuscript has been deemed suitable for publication in PLOS ONE. Congratulations! Your manuscript is now with our production department. 

With kind regards,

on behalf of

Dr. Jaap A. Joles 

Academic Editor

PLOS ONE